# Overexpression of the White Opaque Switching Master Regulator Wor1 Alters Lipid Metabolism and Mitochondrial Function in *Candida albicans*

**DOI:** 10.3390/jof8101028

**Published:** 2022-09-28

**Authors:** Susana Hidalgo-Vico, Josefina Casas, Carolina García, M. Pilar Lillo, Rebeca Alonso-Monge, Elvira Román, Jesús Pla

**Affiliations:** 1Departamento de Microbiología y Parasitología-IRYCIS, Facultad de Farmacia, Universidad Complutense de Madrid, Avda. Ramón y Cajal s/n, 28040 Madrid, Spain; 2Research Unit on BioActive Molecules (RUBAM), Department of Biological Chemistry, Instituto de Química Avanzada de Cataluña, Jordi Girona 18–26, 08034 Barcelona, Spain; 3Departamento de Química Física Biológica, Instituto Química Física “Rocasolano”, Consejo Superior de Investigaciones Científicas, Serrano 119, 28006 Madrid, Spain

**Keywords:** commensalism, gut, lipid, mitochondrial activity, reactive oxygen species, *Candida albicans*, Wor1

## Abstract

*Candida albicans* is a commensal yeast that inhabits the gastrointestinal tract of humans; increased colonization of this yeast in this niche has implicated the master regulator of the white-opaque transition, Wor1, by mechanisms not completely understood. We have addressed the role that this transcription factor has on commensalism by the characterization of strains overexpressing this gene. We show that *WOR1* overexpression causes an alteration of the total lipid content of the fungal cell and significantly alters the composition of structural and reserve molecular species lipids as determined by lipidomic analysis. These cells are hypersensitive to membrane-disturbing agents such as SDS, have increased tolerance to azoles, an augmented number of peroxisomes, and increased phospholipase activity. *WOR1* overexpression also decreases mitochondrial activity and results in altered susceptibility to certain oxidants. All together, these changes reflect drastic alterations in the cellular physiology that facilitate adaptation to the gastrointestinal tract environment.

## 1. Introduction

*C. albicans* is a prominent fungal commensal of the oral cavity and vaginal and gastrointestinal tract of humans. Colonization occurs early after birth and results in as much as 70% of the adult population being colonized without noticeable disease. A currently accepted view is that *C. albicans* colonization in humans is beneficial to the host [1,2] and has a role in promoting intrinsic defense mechanisms of the host against other diseases [3,4]. This fungus can, however, overgrow and/or invade internal tissues and organs, thereby causing diseases that are collectively called candidiasis; candidiasis may be relatively benign (vaginal candidiasis, recurrent in certain women) or the oral thrush of the newborn. However, they can be more severe in individuals with congenital or acquired defects in immunity (e.g., chronic mucocutaneous candidiasis (CMC)) or those with predisposing conditions such as diabetes or long-term antibacterial and anticancer therapy [5,6,7]. Although it is an opportunistic pathogen, this fungus does not have a significant saprophytic life outside the human body compared to other fungi such as *Aspergillus* spp. or *Cryptococcus*, which determines the high incidence of its infections in the community. In fact, while catheters are an important source of fungal nosocomial infections [8], most candidiasis are endogenous as revealed by molecular typing studies [9] and by the intestinal expansion of *Candida* species that precedes its dissemination through the bloodstream in recipients of a hematopoietic cell transplant [10]. Therefore, understanding the factors promoting—or precluding—the establishment of the fungus in the gut is of primary importance for future therapeutical interventions that may not only control its proliferation but reinforce its potential beneficial effects.

Different genes promoting the colonization of *C. albicans* in the gastrointestinal tract have been identified in recent years, mainly using models of commensalism in mice that rely on the reduction in the bacterial microbiota by antibiotic therapy [2,11,12,13]. Among them, the Wor1-Efg1 axis seems to be especially relevant. *EFG1* encodes a morphogenetic regulator that regulates the dimorphic growth characteristic of this yeast [14] in response to environmental conditions. Interestingly, Efg1 is important for adaptation to the gastrointestinal tract and *efg1*∆ mutants have higher fungal loads in the gut [15,16] compared to wild type cells. Among its diverse cellular functions, Efg1 negatively regulates the expression of the master regulator of the white-to-opaque transition *WOR1*, promoting the formation of opaque cells (the mating competent form in this fungus) from standard white cells leading the formation of tetraploids from **a** or α diploid cells [17]. This conversion, called the **wo** transition, is an epigenetic process governed not only by Wor1 [18,19,20,21] but also Czf1, Wor2, Ahr1, and Wor3 [22,23], ultimately forming an intertwined regulatory network controlling the **wo** transition [24,25,26]. Overproduction of Wor1 promotes the conversion of white-to-opaque cells in **a** or α cells but also promotes gut colonization by generating the so-called GUT cells in **a/**α diploids upon travelling through the mammalian tract, while, in contrast, *wor1* cells are rapidly depleted from this niche [27]. GUT cells have a different cell surface appearance compared to opaque cells, with specific protuberances and are stably stained with phloxine B. Opaque cells, however, revert at 37 °C losing the phloxine B staining, and colonize more easily in the skin—but not the gut [27,28]. Transcriptional, proteomic, and phenotypic profiling have revealed important differences in cells overexpressing *WOR1* (WOR1^OE^ cells) versus standard opaque cells [27,29,30], with some pathways being activated (e.g., utilization of fatty acids and N-acetyl glucosamine) and others repressed (e.g., use of 2C sources via the glyoxylate cycle) [27,29,30]. Overproduction of *WOR1* also results in increased adhesion to the gut mucosa and susceptibility to bile salts, a phenotype perhaps not expected for an intestinal commensal microbe [31]. In this work, we show that a major consequence of Wor1 production is a drastic alteration of the quantitative and qualitative lipid composition in the cell as determined by lipidomic analysis, rendering cells with increased tolerance to certain azoles and significantly altering the mitochondrial activity. Therefore, metabolic adaptation is an essential feature for the efficient colonization of this fungus in the mammalian gut.

## 2. Materials and Methods

### 2.1. Strains and Growth Conditions

The strains used in this study are described in Table 1. Yeasts were grown at 37 °C in YPD medium (1% yeast extract, 2% peptone, 2% glucose) unless otherwise specified. When necessary, the growth media was supplemented with 10 µg/mL of doxycycline (dox, AppliChem PanReac, Castellar del Vallès, Barcelona, Spain) 24 h before and during the experiment to regulate the TET expression system on cells. YPG media (2% glycerol) was used for mitochondrial analyses to force respiration. Growth was determined by OD_600_ measurements. To induce the formation of peroxisomes, cells were grown overnight (16 h) on YPD or YNB olive oil/Tween 80 (0.67% yeast nitrogen base without amino acids, 0.5% ammonium sulfate, 0.12% olive oil/0.2% Tween 80 and 20 µg/mL chloramphenicol) and microscopy images were obtained with a Nikon Eclipse TE2000-U microscope equipped with a Nikon B-2E/C filter. Drop tests were performed by spotting drops containing 10^5^ cells and ten-fold serial dilutions of stationary cells onto YPD supplemented with the indicated compounds. Microaerophilia conditions were achieved in an anaerobic chamber with a commercial system (GENBox Microaer, BioMérieux, Madrid, Spain). Susceptibility to amphotericin B was tested in a YPD medium supplemented with different concentrations of the drug in multiwell plates. A total 10^4^ cells from stationary phase cultures were inoculated and plates were incubated at 37 °C. Absorbance readings at 600 nm at 12, 24 and 48 h are represented in a heat map as the median of two biological replicates.

### 2.2. Molecular Biology Procedures and Plasmid Constructions

Fluorescent labeling of peroxisomes was achieved by the construction of an N-terminal GFP fusion to the acyl-CoA oxidase Pxp2 protein. To integrate the construction on the *ADH1* locus, the *Kpn* I restriction site from the *PXP2M* sequence was removed. First, two PCRs were carried out for the amplification of the *PXP2* ORF from the SC5314 strain; the first PCR amplified a 1274 pb fragment from the ORF (without the ATG) to the *Kpn* I site by using primers o1-PXP2 (which incorporates a *Not* I sequence) and o2-PXP2. The second PCR amplified a 948 pb fragment from the *Kpn* I site by using primers o3-PXP2 and o4-PXP2 (that incorporates a *Bgl* ll restriction site at the end of the sequence). Both products were used to generate a 2197 pb *PXP2M* (M, modified) fragment by performing an overlapping PCR using o1-PXP2 and o4-PXP2 primers. The product was inserted on the pGEM-T (Promega, Alcobendas, Madrid, Spain) vector obtaining the pGEM-T-PXP2M plasmid. Second, we constructed a vector that permits the expression of the fused protein following these steps; (1) the GFP-myc fragment was obtained from pDHM0-GFP [36] digested with *Sal* I y *Not* I and inserted into the plasmid pAS0-RFP [35] digested with the same enzymes, generating the pAS0-GFP-myc vector; (2) the pGEM-T-PXP2M plasmid was digested with *Bgl* II and *Not* I to obtain the *PXP2M* fragment and inserted into the digested pAS0-GFP-myc (Román, unpublished) to obtain the pAS0-GFP-PXP2M plasmid; (3) the *OP4* promoter was replaced by the constitutive and strong promoter *TDH3*; for this purpose, the *TDH3* promoter was obtained from plasmid pNIM1RX-RFP [36] as a *Spe* I/*Xho* I fragment that was inserted in pAS0-GFP-PXP2M generating pAS3-GFP-PXP2M. (4) Finally, the *ADH1* integration regions in this vector were replaced by the flanking regions of *ARD1*; for this purpose, the TDH3-GFP-PXP2M fragment was excised as a *Nsi* I/*Spe* I fragment and inserted into the pDARD1 vector [37] that was also digested with these enzymes, generating the final vector pDS3-GFP-PXP2M. Strains were obtained by performing electroporation with a *Kpn* I-*Sac* I fragment and the transformants were selected on YPD media supplemented with 200 µg/mL of nourseothricin. A list of primer sequences is given in Table 2.

### 2.3. RNA Isolation and RT-qPCR Analysis

Cells were grown overnight (16–18 h) in YPD and diluted to an optical density (O.D.) of 0.1 in the same medium; cells were allowed to grow to an O.D. = 0.8 and collected with low-speed centrifugation (5 min, 10,000 rpm). RNA was extracted using nitric acid-treated glass beads in a Fastprep breaker and purified using the RNeasy kit (Qiagen, Las Rozas, Madrid, Spain) according to the manufacturer’s protocol. A total of 1 µg of RNA was used to synthesize cDNA by using the PrimeScript RT kit (Takara, Torrejon de Ardoz, Madrid, Spain) and reactions were prepared using the SYBR-Green PCR Master Mix (Roche, Madrid, Spain). Real-time qPCR assays were performed using a QuantStudio 7 Real-Time PCR System (Roche Diagnostics, San Cugat del Vallès, Barcelona, Spain). Primer sequences are listed on Table 2. The expression of each gene was normalized by relativizing the values to the housekeeping gene *ACT1* and the relative expression ratios were obtained as described by Livak and Schimittgen [39].

### 2.4. Mitochondrial Function Assays

The mitochondrial membrane potential was monitored by performing JC-1 dye staining (Invitrogen, Alcobendas, Madrid, Spain). A total of 10^6^ cells from overnight cultures in YPG were transferred to 1 mL of fresh prewarmed YPG media. JC-1 was added to 1.5 μM and cells were incubated for 30 min at 37 °C in the dark before being washed with PBS (3×). Cells were analyzed by flow cytometry using a FACScan (Becton Dickinson, San Agustín del Guadalix, Madrid, Spain) cytometer at an excitation λ of 488 nm and detected in both FL1 and FL2 channels. The ratio in red/green (FL2/FL1) was used as a measurement of mitochondrial functionality. Sensitivity to chloramphenicol (Sigma, Madrid, Spain) was determined by performing drop tests on YPD or YPG plates supplemented with 2 or 4 mg/mL of this antibiotic. Plates were further incubated at 37 °C for 48 h before being scanned.

### 2.5. Protein and Immunoblot Methods

All procedures involving proteins were as previously described [40]. Protein extracts were equalized by absorbances at 280 nm and loaded proteins for western blot analyses were probed with anti-Phospho-p38 MAPK (Thr180/Tyr182) (Cell Signalling Technology, The Netherlands) and anti-Actin clone C4 (MP Biomedicals, Madrid, Spain) antibodies. Membranes were developed with the Odyssey system (LI-COR) 2.6.

### 2.6. Analysis of Lipid Content

Nile red staining was performed on stationary cells incubated on YPD +/− dox. Briefly, 2.5 × 10^6^ cells were washed with PBS and stained with 5 µg/mL of Nile red (Sigma) and 1 µg/mL of DAPI (Sigma) when needed for 10 min on ice. As a positive control, *hog1* cells incubated overnight in YPD NaCl 1 M were used, as this mutant accumulates lipid droplets when cells are exposed to osmotic stress [41]. Cells were washed three times with PBS and fixed on 4% formaldehyde before microscopic examination using a Nikon B-2E/C filter (Nile Red) or Nikon UV-2A (DAPI). Cells were also analyzed by performing flow cytometry using a Guava easyCyte (Millipore, Darmstadt, Germany) cytometer at an excitation λ of 488 nm.

The quantitative lipid analysis was carried out by the lipidomics service of the Research Unit on Bioactive Molecules (RUBAM) at the Institute of Advanced Chemistry of Catalonia (IQAC/CSIC). The extraction was carried out using a protocol previously described [42] with some modifications. For the extraction of samples, we started from overnight grown cells and 25 optical density units were collected per strain. Cells were centrifuged for 5 min at 4 °C and the supernatant was removed entirely before resuspension in 500 μL of methanol/0.01% BHT. Glass beads were used for mechanical cell disruption in FastPrep for 30 s and 8 cycles at 5.5 m/s with 30 s on ice in between. To obtain the supernatant, tubes were punctured at the base with a flame-heated needle and mounted on top of cold ones to remove the beads. The samples were transferred to glass vials for cold evaporation under N_2_ flow and stored at room temperature. For the analysis of phospholipids and non-polar lipids, the dried residue was resuspended in 100 μL of H_2_O, 750 μL of 2:1 methanol/chloroform, and the internal standards (Avanti Polar Lipids, Alabaster, AL, USA) and transferred to a new vial. The sample was incubated overnight at 48 °C under stirring in a thermostatic bath. Internal standards were added in the following quantities: 200 pmol C17:0 sphinganine, 200 pmol C17 sphinganine-1-phosphate, 200 pmol C12 ceramide, 200 pmol d18:1/12:0 ceramide, 200 pmol d18:1/12:0 sphingomyelin, 200 pmol d18:1/12:0 hexosylceramide, 500 pmol d18:1/17:0 ceramide trihexoside, 200 pmol 16:0/18:1 phosphatidylcholine (PC), 100 pmol 17:0 lysophosphatidylcholine (LPC), 133 pmol 16:018:1 lysophosphatidylethanolamine (LPE), 200 pmol 17:1 LPE, 123 pmol 16:0/18:1 phophatidylserine (PS), 200 pmol 17:1 lysophosphatidylserine (LPS), 200 pmol 16:0/18:1 glycerophosphoglycerol, 200 pmol 17:1 lysoglycerophosphoglycerol, 145 pmol 17:0 monoacylglycerol (MAG), 166 pmol 17:0/0:0/17:0 diacylglycerol (DAG), 200 pmol 17:0/17:0/17:0 triacylglycerol (TAG), 156 pmol 17:0 cholesteryl ester, 4000 pmol stigmasterol, and 200 pmol C16 D3 carnitine. The contents were evaporated in a SpeedVac concentrator and stored at −80 °C. The dried residues for sphingolipids analysis were prepared as follows. A total of 750 μL of a 2:1 methanol/chloroform solution and the internal standards were added and incubated overnight at 48 °C. The samples were tempered, and 75 μL of 1 M KOH in methanol was added and incubated at 37 °C for 2 h. A total of 75 μL of 1 M acetic acid was added, and the samples were evaporated to a dry residue and stored at −80 °C. All samples were resuspended in 200 μL of methanol, centrifuged at 9300 rcf for 3 min, and 130 μL of the supernatants was transferred to a vial of UPLC for analysis. Lipids were analyzed by performing liquid chromatography coupled with a high-resolution mass spectrometer (LC-HRMS) using an Acquity Ultra High-performance Liquid Chromatography (UHPLC) System connected to a TOF (Time of Flight) LCT Premier XE detector according to previously described protocols [43,44] which were minimally modified. The extracts were injected in an Acquity UHPLC BEH C8 column (1.7 μm particle size, 100 mm × 2.11 mm, Waters, Cerdanyola del Vallès, Barcelona, Spain) at a flow rate of 0.3 mL/min. The mobile phases used were (A) 2 mM ammonium formate in H_2_O and (B) 0.2% formic acid depending on the gradient.

## 3. Results

### 3.1. Lipid Content Is Altered in WOR1^OE^ Strain

In a previous study, we reported significant proteome changes in a strain overexpressing the **wo** master regulator *WOR1* (named as CAI4-WOR1^OE^ or WOR1^OE^ strain) [30]. WOR1 mRNA in this strain is 4× the parental controls (not shown). We noticed the presence of several enzymes involved in lipid biosynthesis/storage and peroxisome function in differentially abundant proteins when comparing WOR1^OE^ to the control strain carrying an empty plasmid, CAI4-pNRUe. Some of these proteins were only identified in the WOR1^OE^ strain, such as Lro1, involved in triglyceride biosynthesis, or Ole2 and Fad2 fatty acid desaturases; others were decreased, such as Acp1, Smp2 or Lag1, involved in phospholipid and fatty acid synthesis. We thus used the lipophilic dye Nile red to quantify neutral as well as membrane lipids in stationary phase growing cells by performing flow cytometry. The mean fluorescence intensity in the FL2-yellow channel (NR FL2-H) is shown in Figure 1A.

A significant increase in the fluorescence was detected in the WOR1^OE^ strain compared to the wild type reference (MFI, 193 vs. 39.6). This was also made evident by fluorescence microscopy (Figure 1B). The staining of wild type cells with Nile red and DAPI revealed a diffuse intracellular cytoplasmic pattern (red) external to the nucleus (green) (Figure 1C). The experiment was performed in the presence of doxycycline; in this case, both the CAI4-pNRUe and CAI4-WOR1^OE^ strains displayed a similar fluorescence level (Figure 1D) confirming that alterations in lipid storage were due to *WOR1* overexpression. We have previously shown that the HOG MAPK pathway is involved in lipid homeostasis in *C. albicans* upon osmotic challenge [41] and that WOR1^OE^ restores the deficient colonization of *hog1* mutants in the mouse gut (Román, submitted), suggesting that *WOR1* and *HOG1* may play complementary roles in cellular physiology. We tested this assumption by constructing a *WOR1* overproducing strain in a *hog1* background. Lipid accumulation caused by WOR1^OE^ was severely diminished in this background (337 ± 37 versus 76 ± 13 in wild type background, 126 ± 27 versus 62 ± 5 in *hog1* background) (Figure 1D,E).

We next determined the lipid content profile of fungal cells. For this purpose, cells were disrupted mechanically, and the total lipid content was extracted in methanol; the debris was treated differently for each lipid species and analyzed with an LC-HRMS chromatograph connected to a TOF detector. The main results are expressed as the ratio of amounts (in pmol/unit of O.D.) equivalent to each internal standard between both strains (CAI4-WOR1^OE^/CAI4-pNRUe). We observed the following trends (Figure 2 and Appendix A).

Total triacylglycerols (TAG) showed a 1.3 ratio, with increases (≥2) in species C46, 48, 50, 51, 54, and 56 and no molecular species being less abundant. Regarding diacylglicerols (DAG), the ratio was higher (1.6) with six species showing a significant increase (C32, 34, 36, and 38); in particular, species C38:1 was found in a ratio = 4.8; again, no molecular species were found to be less abundant than the control cells. WOR1^OE^ cells showed an increase in the content of lysophosphatidylethanolamines (LPE) and lysophosphatidylcholines (LPC) (ratios 2.7 and 2.4); two molecular species showed ratios of 4.7 (C16:0) and 7.1 (C18:0) and three LPC had ratios of 4.4 (C16:0), 3 (C18:0) and 2.2 (C18:1). Changes were also observed in two phosphatidylinositol (PI) species (C34:0, ratio 2.3, and C36:2, ratio 3), but not in total content (ratio 1.3). The whole content of phosphatidylserines (PS) or phosphatidylcholines (PC) was equivalent, conversely, to ratios of 0.4 and 0.6, respectively. For PS, all molecular species were found to be reduced compared to the control except one. For PC, molecular species that were decreased in WOR1^OE^ were those presenting a higher number of carbons (C38:2 to C42) while no differences were found for species C32 to C38:1. We also observed changes between sphingolipid species, with a ratio of 1.6 for sphinganine and 0.6 for phytosphingosines. No major changes were found between both strains in total dihydroceramides content (ratio 1.2), phytoceramides (1) and ceramides phosphoinositol (1). Only one dihydroceramide (d18:0/18:0, ratio 4.6) and two of the ceramides phosphoinositol (t42:0/d42:0, ratio 2.8, and t44:0/d44:0, ratio 2.7) were found in higher amounts. No significant differences were observed in the ergosterol content for WOR1^OE^ cells (ratio 1.2). In summary, lipid content becomes altered in overexpressing *WOR1* cells with an increase in neutral lipids detected by Nile red and LC-HRMS chromatography and clear alteration of cellular lipids, in particular lysophosphatidylcholine and lysophosphatidylethanolamine.

### 3.2. WOR1 Overexpression Enhances the Utilization of Phospholipids and Increases the Number and Size of Peroxisomes

Peroxisomes are organelles involved in the β-oxidation of fatty acids and the synthesis of membrane lipids. To determine if the structural changes observed correlate with alterations in peroxisome biogenesis, peroxisomes were fluorescently labelled by integrating GFP-PXP2M gene fusion. Pxp2 encodes an acyl-CoA oxidase that participates in the β-oxidation of fatty acids that is specifically localized in the peroxisomes. Peroxisomes were then observed under fluorescence microscopy in cells that were incubated in YPD or YNB medium supplemented with olive oil/Tween 8 as a carbon source, which has been reported to increase their synthesis [45]. As shown (Figure 3), *WOR1* overexpression also increased the number of peroxisomes compared to the control strain when grown in both media.

### 3.3. WOR1 Overexpression Alters the Susceptibility to Certain Membrane Disturbing Compounds

Since alteration in the composition of the cellular lipids was detected in the WOR1^OE^ strain, we explored the susceptibility to compounds that disturb membrane integrity in a standard drop assay on YPD plates supplemented with SDS which acts as a detergent (Figure 4A).

The overexpression of *WOR1* rendered cells sensitive to SDS. This phenotype can be associated with an altered membrane composition and could explain the sensibility to bile salts previously reported [31]. We also tested the susceptibility to fluconazole and miconazole, since these antifungals prevent ergosterol synthesis by blocking the 14-α-lanosterol demethylase, encoded by the *ERG11* gene. Spot assays showed that WOR1^OE^ cells are more tolerant to these compounds compared to the control strain (Figure 4B). This phenotype was observed either in normoxia or microaerophilia conditions. We also quantified *ERG11* expression with RT -qPCR, which showed a ≈ 50% reduction in mRNA (Figure 4C). This lower level of expression did not correlate with a significant variation in the total ergosterol content. Interestingly, WOR1^OE^ cells were found to be more resistant to amphotericin B, a polyene drug, in a standard liquid assay (Figure 4D), reinforcing the idea that these cells have an altered membrane lipid organization.

### 3.4. Mitochondrial Function Is Compromised by WOR1 Overexpression

We next checked the sensitivity to different oxidants on YPD plates supplemented with different compounds (Figure 5A). *WOR1* overexpression resulted in sensitivity to 2.5 mM diamide, 0.2 mM menadione, and 5 mM H_2_O_2_ compared to the control strain. Interestingly, cells were not sensitive to arsenic-derived compounds, such as arsenite (As III) and arsenate (As V), which have also been shown to generate intracellular ROS and increase the mitochondrial membrane potential in *C. albicans* [46].

As oxidative stress is sensed in *C. albicans* by the HOG MAPK pathway [9], we explored if signaling through this pathway was altered due to *WOR1* overexpression. Cells were exposed to 3 mM H_2_O_2,_ and samples were taken at different time points to determine the activation of Hog1 by performing Western blot. In the wild-type strain, Hog1 became phosphorylated in exponentially growing cells (at O.D. = 0.8) at 30 s after the challenge; in contrast, WOR1^OE^ cells were delayed in this response, and it took 5 min to reach similar phosphorylation levels (Figure 5B). These data, together with the lipidomic results, led us to hypothesize that mitochondrial functionality could be altered. We wondered whether these putative mitochondrial alterations could be revealed by using chloramphenicol, which is an antibiotic that inhibits bacterial protein synthesis, but interferes with eukaryotic mitochondrial function at high concentrations. As it is shown in Figure 5C, *WOR1* overexpressing cells (but not control cells) are highly sensitive to this drug, reinforcing the idea that mitochondria are altered in WOR1^OE^ cells. This sensitivity was observed either using glucose (fermentable) or glycerol (non-fermentable) as a carbon source (data not shown). To further characterize the defect present in WOR1^OE^ cells, we used the cationic JC-1 fluorochrome; this fluorochrome accumulates in mitochondria and aggregates, resulting in red fluorescence emission for normally polarized mitochondria while remaining green for depolarized mitochondria. Therefore, high ratios of both signals (red/green; FL2-H/FL1-H) are indicative of a functional organelle under the conditions tested. As shown in Figure 5D, wild type cells in YPG represented a rather homogenous population, with FL2-H/FL1-H ratio ≈ 6 (FL2-H Mean Fluorescence = 2471, FL1-H Mean Fluorescence = 409) indicative of active mitochondria. However, cells overexpressing *WOR1* showed a drastic decrease in this ratio (FL2-H/FL1-H ratio ≈ 0.4; FL2-H Mean Fluorescence = 333, FL1-H Mean Fluorescence = 816) indicative of much less-active mitochondria. Altogether, these data indicate that *WOR1* overexpression produces mitochondrial functional alterations that are responsible, at least in part, for altered susceptibility and response to certain oxidants.

## 4. Discussion

The master regulator of the **wo** switching *WOR1* is important for *C. albicans* colonization in the gastrointestinal tract [27,31]. In this study, we analyzed the role of Wor1 in lipid homeostasis, mitochondrial function, azole susceptibility, and response to stress by comparing *WOR1* overexpressing cells with their isogenic parental strain. We present here a connection between Wor1 overproduction and lipid metabolism, showing that WOR1^OE^ cells increase the number of lipid droplets and neutral lipids by measuring fluorescence with Nile red staining (Figure 1). Although lipidomic analyses have not shown differences in the total content of triacylglycerols (TAGs), we previously identified Lro1 as a unique protein in WOR1^OE^ cells compared to the parental control strain, which uses diacylglycerol (DAG) to synthesize TAGs for storage in lipid droplets that could explain the increase in these structures in this strain. This suggests that cells that overproduce Wor1 may have decreased energy requirements, consequently storing energy in triglycerides because of an impaired mitochondrial function (see later). The decreased levels in phosphatidylserine (PS) could also explain some of the phenotypes we observed in WOR1^OE^ cells. Yeast cells synthesize PS from CDP-DAG and serine through the Cho1 enzyme, while Psd1 and Psd2 (PS decarboxylases in mitochondria and endosomes, respectively) convert PS to PE. Deletion of *CHO1* led to alterations in total lipid composition, filamentation and cell wall structure, increased SDS and oxidants susceptibility and diminished virulence in a model of systemic candidiasis in mice [47,48,49,50]. These phenotypes are shared with WOR1^OE^ cells (this work and Román, submitted). The increase in the amount of lysophospholipids suggests increased activity of phospholipase, which is found to be higher in WOR1^OE^ cells on solid MEA and SEA-specific media (Román et al., submitted).

We also show that Wor1 overproduction results in increased resistance to fluconazole and miconazole and that this phenotype is more evident under a microaerophilic environment. Both antifungals act by inhibiting the 14-α-lanosterol demethylase encoded by *ERG11* which leads to a decrease in ergosterol levels, altering the plasma membrane, and an accumulation of intermediary metabolites that are toxic to the cell [51]. Under hypoxia, a condition that *Candida* cells encounter in the gut, *ERG11* transcript levels and the level of other factors related to the ergosterol biosynthesis, such as *ERG3* and *ERG5*, become increased [52]; however, our data demonstrate that this increased resistance is not dependent on *ERG11* overexpression since mRNA levels were, in fact, lower in WOR1^OE^ cells compared to the control cells and the total levels of ergosterol in our lipidomic analysis were similar in both strains. Azole resistance could also be acquired by overexpression of efflux pumps such as MDR and CDR that lead to a reduced level of the antifungal inside the cell [53]. *WOR1* diminishes *EFG1,* which positively regulates *CDR* gene expression [54] and, in agreement, *efg1* mutants display enhanced susceptibility to azoles [55]. These phenotypes do not fit with the increased tolerance of Wor1 overproducing cells; however, although we do not have data regarding the impact of Wor1 on their expression, Nile red data do not support the CDR increase as responsible for this phenotype since this compound is a substrate of the major transporters responsible for azole resistance in *C. albicans* (Cdr1, Cdr2, and Mdr1) and is efficiently exported out of the cell [56].

Azole resistance has also been linked to mitochondrial function. Mitochondria are required for the synthesis of cardiolipin and phosphatidydiylglycerol, specific phospholipids found in mitochondrial membranes, and mutations in mitochondrial biosynthetic genes alter the functionality of respiratory complexes. Conversely, in *S. cerevisiae* and *C. glabrata,* the loss of mitochondrial function in petite mutants results in resistance due to the upregulation of ABC transporters [57], in *C. albicans* or *Cryptococcus* spp., these petite negative elements cannot survive without mtDNA. Both susceptible and resistant mitochondrial mutants have been documented [58,59]. Our results suggest that the overexpression of *WOR1* induces an alteration in mitochondria functionality; WOR1^OE^ cells have increased non-functional mitochondria (Figure 5), as seen by JC-1 staining under respiratory conditions, and increased susceptibility to chloramphenicol, which is a protein inhibitor that has been shown to interfere with the eukaryotic mitochondrial function at high concentrations [60]. A reduced mitochondrial functionality can also explain susceptibility to azoles, as we have previously shown how mutants defective in the Fzo1 GTPase that mediates mitochondrial fusion, show reduced efflux activity of CDR pumps and an altered response to oxidants that are mediated, at least in part, by the Hog1 MAP kinase as occurs here [61,62]. *hog1* mutants also have a defective respiratory metabolism [63] as occurs with WOR1^OE^ cells which have increased sensitivity to sodium azide [31], which reinforces the connection between mitochondria and *WOR1*-mediated changes. As phospholipid synthesis involves both the endoplasmic reticulum (ER) and the mitochondria, we anticipate a significant ER stress in WOR1^OE^ cells which could explain some of the cell wall phenotypes of these cells such as sensitivity to Congo red (data not shown). The azole resistance phenotype is consistent with alterations in the lipid profile, as changes in lipids have been associated with azole resistance [64,65]. However, it should be noted that one consequence of *WOR1* overexpression is diminishing levels of *EFG1*, whose deletion has been previously reported to cause sensitivity to certain azoles [55], pointing towards *EFG1*-independent mechanisms controlling azole resistance as it has been shown by the analysis of *wor1 efg1* mutants [66]. In any case, the increased tolerance to amphotericin B indicates that lipid profile modifications also affect the plasma membrane.

What do all these changes tell us about *C. albicans* colonization in the gut? Adaptation to this niche would in theory require a reduced dependence on certain carbon sources and oxygen availability. Glucose is scarce at this location, but other carbon sources may be more abundant. An alteration in lipid metabolism was already observed by Pande in their transcriptome analysis [27,31]. Interestingly, *C. albicans* colonization results in increased resistance to *C. difficile* infection, at least in part due to the changes in the molecular lipid species of the cecum of antibiotic-treated mice mediated with augmented unsaturated fatty acids [67]. This suggests that *C. albicans* alters the luminal intestinal composition to induce a more favorable metabolic environment for its growth and Wor1 could mediate this process. The gastrointestinal tract, especially in its distal portions, has a limited availability of oxygen [67] and successful adaptation to this niche should therefore not depend on an extensive oxidative metabolism. WOR1^OE^ cells in mice co-colonized with a wt strain settle preferentially in the cecum and large intestine with lower oxygen concentration [27,31], suggesting that certain mitochondrial functions are not essential for the adaptation to the murine gut. We must also indicate that the biological significance of Wor1 in humans has not been yet addressed. Soll’s group demonstrated that around half of the 27 clinical a/α *Candida* isolates can be induced to switch in vitro and that this effect can be due to mutations in *EFG1* but also due to mutations in other genes [68]. Future studies with clinical isolates may help to rule out the biological significance of *WOR1* overexpression in the human gut. Collectively, our work demonstrates that a major consequence of Wor1 overexpression in *C. albicans* is alteration of the metabolism of the cell connecting mitochondrial function and lipid metabolism with its previously described role in gut colonization.

## Figures and Tables

**Figure 1 jof-08-01028-f001:**
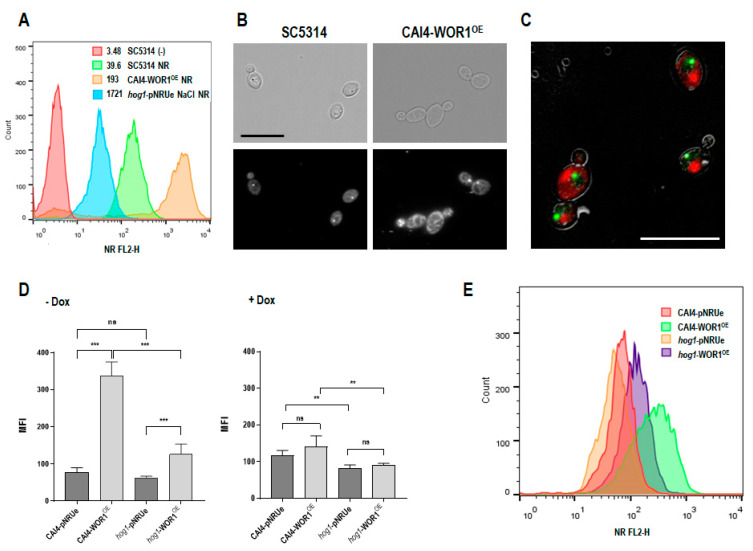
Nile red staining. (**A**) Cells in stationary phase of growth were collected and stained with 5 µg/mL of Nile red (NR). The *hog1* mutant with the empty vector (*hog1*-pNRUe) grown on YPD NaCl 1 M was used as a positive control. Stained cells were analyzed by performing flow cytometry as described in the Material and Methods (λex of 488 nm) and the mean fluorescence intensity (MFI) of a representative experiment is shown as a histogram. The inner legend lists the MFI for each strain analyzed; an unstained control is included (SC5314 (-), red). (**B**) Nile red stained cells of the indicated strains under fluorescence microscopy are shown. The scale bar represents 10 µm cell size. (**C**) Overnight growing CAI4-WOR1^OE^ cells were stained with Nile red (red signal) and DAPI (green signal) for 10 min. The image represents an overlay of the bright field and fluorescent signals. (**D**) Cells were grown in YPD supplemented with or without doxycycline for 24 h and stained with Nile red. Graphs represent the MFI ± SD of five experiments ns: not significant, ** *p* < 0.01, *** *p* < 0.001. (**E**) A representative histogram of samples in D grown in the absence of doxycycline is shown.

**Figure 2 jof-08-01028-f002:**
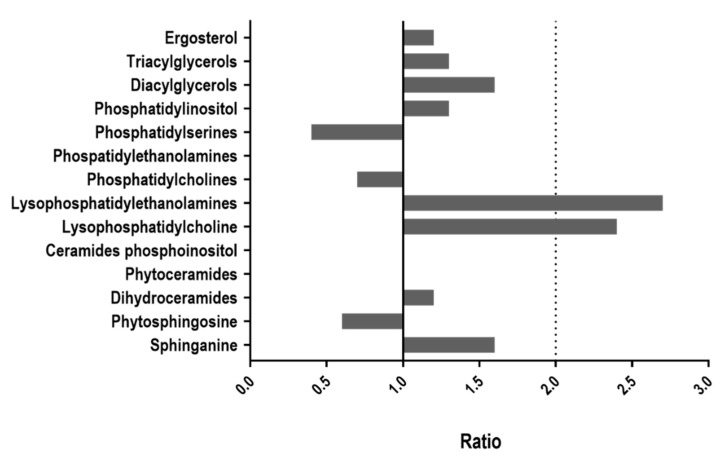
Lipid content quantification by LC-HRMS-TOF. Stationary phase cells were mechanically disrupted in MeOH + BHT and the dry residue was extracted for further analysis in the chromatograph. The total lipid content of each type was calculated with the sum of all the molecular species found in pmol eq./unit of O.D. and expressed in a ratio CAI4-WOR1^OE^/CAI4-pNRUe.

**Figure 3 jof-08-01028-f003:**
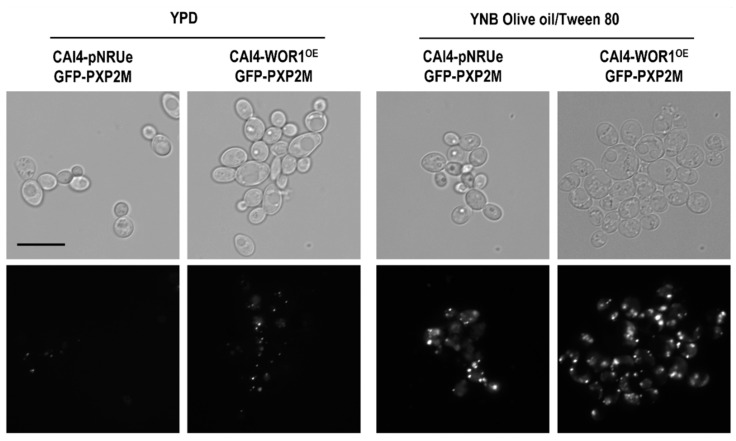
Fluorescent labeling of peroxisomes. CAI4-pNRUe and CAI4-WOR1^OE^ strains producing the GFP-Pxp2M fusion protein were grown 72 h at 37 °C on YPD or YNB supplemented with 0.12% olive oil/0.2% Tween 80 and microscopy images were taken. The scale bar corresponds to a 10 µm size.

**Figure 4 jof-08-01028-f004:**
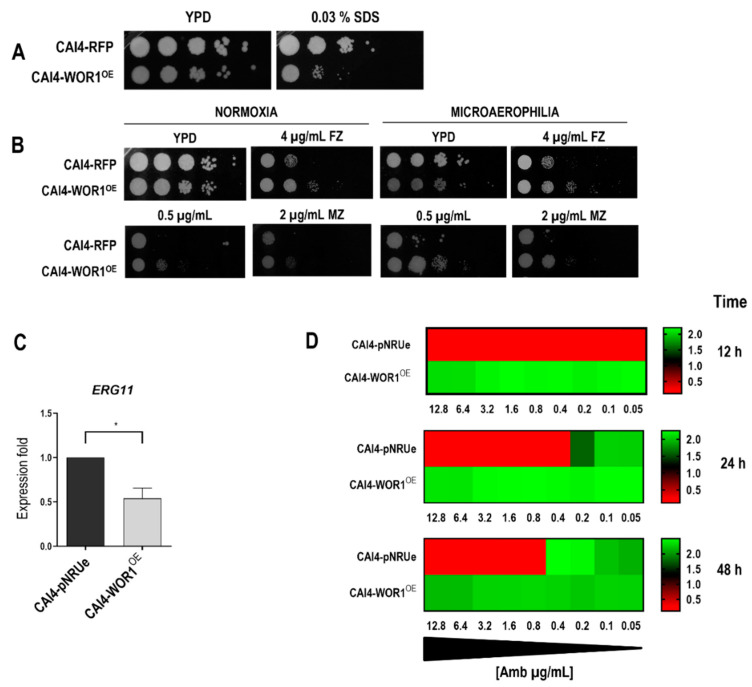
Effect of *WOR1* overexpression on membrane disturbing agents’ sensitivity. Suspensions from the indicated strains grown overnight in YPD at 37 °C were prepared at 2 × 10^7^ cells/mL and five microliters from tenfold dilutions were spotted on YPD plates supplemented with SDS (**A**), fluconazole (FZ), or miconazole (MZ) at the indicated concentrations (**B**). Plates were incubated in normoxia for 24 h or microaerophilia for 48 h before being scanned. (**C**) *ERG11* mRNA transcript expression levels in exponentially growing cells normalized to *ACT1* expression. Fold expression using as a reference the value of the strain, CAI4-pNRUe is shown as the mean ± SD of three independent experiments. * *p* < 0.05. (**D**) Heat map depicts the median of absorbances at the indicated times of CAI4-WOR1^OE^ and control CAI4-pNRUe cells grown in YPD media supplemented with amphotericin B (AmB) at different concentrations (µg/mL).

**Figure 5 jof-08-01028-f005:**
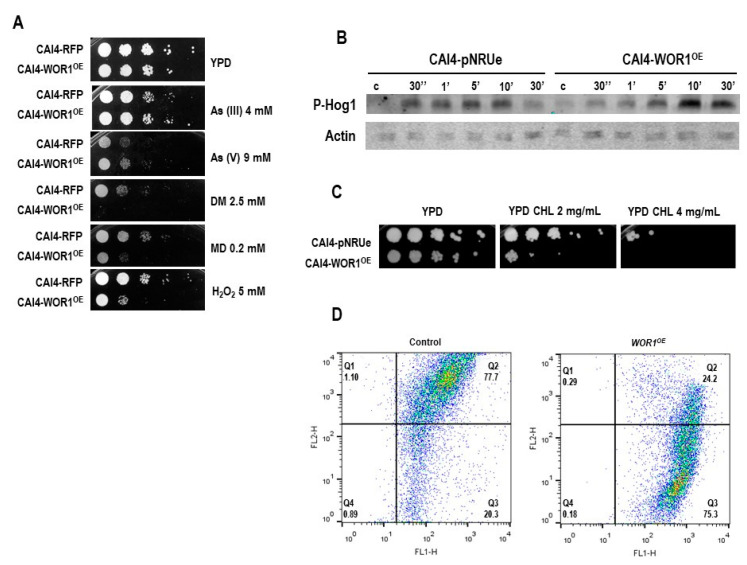
Role of *WOR1* overexpression in the oxidative stress response and mitochondrial function. (**A**) Susceptibility to arsenic-derived compounds and oxidants. 10^5^ stationary cells and tenfold dilutions from either RFP and WOR1^OE^ strains were spotted onto YPD plates supplemented with As (III), As (IV), diamide (DM), menadione (MD) or H_2_O_2_ at the indicated concentrations. Plates were incubated at 37 °C for 24 h in normoxia. (**B**) Exponentially growing cells were exposed to 3 mM H_2_O_2_ and samples were collected before and at 30 s and 1, 5, 10, and 30 min after exposure. Hog1 phosphorylation was detected by western blot using the anti-phospho-Hog1 antibody, and actin was used as a loading control. (**C**) Growth in presence of chloramphenicol. Cells were spotted onto YPD plates supplemented with 2 or 4 mg/mL of chloramphenicol and incubated at 37 °C for 2 days. (**D**) Dot plot analysis of the mitochondrial membrane potential determined by JC-1 dye. 10^6^ cells from CAI4-pNRUe and CAI4-WOR1^OE^ strains grown in YPG were stained with 1.5 µM JC-1 for 30 min at 37 °C and in the absence of light and analyzed by flow cytometry at an excitation λ of 488 nm. FL2-H and FL1-H represent red and green fluorescence, respectively.

**Table 1 jof-08-01028-t001:** Strains of *Candida albicans* used in this work.

Strain Name (Original)	Background Strain and Genotype	Reference
SC5314	Clinical isolate	[32]
CAI4	ura3Δ::imm434/ura3Δ::imm434 iro1/iro1::imm434	[33]
CAI4-pNRUe (REP40)	[CAI4] *ADH1/adh1::tTATET^PR^-myc*-*URA3*	[34]
CAI4-RFP	[CAI4] *ADH1*/*adh1*::*TDH3^PR^tTATET^PR^*-*dTOM2-URA3*	[31]
CAI4-WOR1^OE^	[CAI4] *ADH1/adh1::TDH3^PR^tTATET^PR^-WOR1-myc-URA3*	[31]
SU6	[CAI4-pNRUe] *ARD1/ard1::TDH3^PR^-GFP-PXP2M SAT1*	This study
SU7	[CAI4-WOR1^OE^] *ARD1/ard1::TDH3^PR^-GFP-PXP2M SAT1*	This study
*hog1 ura3* (HI7)	[CAI4] *hog1::hisG/hog1::hisG*	[35]
*hog1*-pNRUe	[*hog1 ura3*] *ADH1/adh1::tTATET^PR^-myc-URA3*	Román, submitted
*hog1*-WOR1^OE^	[*hog1 ura3*] *ADH1/adh1::TDH3^PR^tTATeT^PR^-WOR1-myc-URA3*	Román, submitted

**Table 2 jof-08-01028-t002:** List of primers used in this study.

Primer	Sequence (5′–3′)	Reference
o-ACTQTup	TGGTGGTTCTATCTTGGCTTCA	[38]
o-ACTQTlw	ATCCACATTTGTTGGAAAGTAGA	[38]
o-ERG11up-QT	TTACCTCATTATTGGAGACGTGATG	This study
o-ERG11lw-QT	CACCACGTTCTCTTCTCAGTTTAATT	This study
o1-PXP2	AATGCGGCCGCATGGCTATGCTTACTAAATCTATACATGATG	This study
o2-PXP2	ACATGGATTTGGTCCCATTGATGGTG	This study
o3-PXP2	CACCATCAATGGGACCAAATCCATGT	This study
o4-PXP2	GAAGCAGCTGCTAAATTATCAAGATAATTAGATCTAAT	This study

## Data Availability

Not applicable.

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
