# Peer review of "Overexpression of the White Opaque Switching Master Regulator Wor1 Alters Lipid Metabolism and Mitochondrial Function in Candida albicans"

_jof, 2022, doi:10.3390/jof8101028_

Round 1
Reviewer 1 Report
Overall impression
The authors present a phenotypic characterization of a Candida albicans strain overexpressing a key regulator of morphology, WOR1. Wor1 is involved in the transition from white to opaque colony phenotype and in the adaption of the commensal fungus to life in the human digestive system (the so-called GUIT morphology). The authors report that overexpression of WOR1 leads to loss of mitochondrial activity, lipid droplet accumulation, change of lipid composition and changes in sensitivity to antifungal and antibacterial agents. The experiments are well-designed, well-executed and well-described. The introduction and discussion are accurate, informative and concise, reflecting the strong track record of the authors in the field of Candida albicans biology. Citations are appropriate. While the paper has a few shortcomings in the discussion of the results, the findings are solid (if somewhat preliminary) and worthy of publication.
Major points of critique
The main weakness of the manuscript lies within its descriptive nature. While each of the findings is well documented, it is not clear how the observations connect or what the underlying mechanism is. It is understood that finding and describing the mechanism explaining the WOR1-OE phenotype may take years of work and goes beyond the scope of this manuscript. But since much of the discussion is somewhat speculative, this reviewer will offer counterspeculations and additional considerations to improve the quality of manuscript.
1. Lipid profile (Figure 2). The figure shows a striking observation in membrane lipid composition that needs to be discussed. Lysophospholipids are produced from the corresponding phospholipids through the action of phospholipase A (PLA). The finding that phosphatidylserines/phosphatidylcholines decline while their respective lyso-forms increase points to WOR1-mediated activation of PLA. If that were true, it would greatly increase our understanding of WOR1-mediated stress responses.
2. Lipid droplets (Discussion, first paragraph). The authors document an increase in lipid droplets in WOR1 overexpressing cells and speculate that this might be due to the strain’s higher energy needs. This reviewer would like to offer two countertheories. First, an increase in lipid storage can be seen as a reflection of decreased energy requirements as cells are able to store more of their metabolic energy in triglyceride form. Second, lipid metabolism requires mitochondrial function and an increase in unmetabolized lipids can be seen as a consequence of impaired mitochondrial activity.
3. Overall health of the WOR1-OE strain. Figures 1 and 3 suggest that the WOR1-OE strain might be experiencing serious stress, as evidenced by large vacuoles and bloated appearance of cells in light microscopy. This begs the question of how WOR1 overexpression affects growth and viability. Could the authors provide data or references on the OE-strain’s generation time?
4. Ergosterol content. The authors show elevated ergosterol content in WOR1-OE cells and, after ruling out some explanations, suggest that this might be due to some unspecified mitochondrial influence. This reviewer is left to wonder if there is indeed elevated ergosterol in the plasma membrane or if the observation can be explained by an increase in cell size or intracellular membranes. Could the authors briefly test the susceptibility of WOR1-OE cells to amphotericin to establish if plasma membrane composition is altered?
Minor points/editorials
1. WOR1 overexpression is associated with increased azole tolerance. However, line 76 says otherwise (increased susceptibility)
2. There are a few grammatical problems that should be fixed during copy editing of the manuscript.
Reviewer 2 Report
The manuscript entitled "Overexpression of the white opaque switching master regulator Wor1 alters lipid metabolism and mitochondrial function in Candida albicans", co-authored by Hidalgo-Vico and co-authors, describes that WOR1 overexpression causes changes in total lipid content and the composition of structural and reserve molecular lipids. The authors also provide evidence that cells overexpressing Wor1 are hypersensitive to membrane disturbing agents like SDS, showing increased tolerance to azoles and an augmented number of peroxisomesy. Thea authors also show that WOR1 overexpression led to a decreased mitochondrial activity and an increased susceptibility to the oxidants diamide (DM), menadione (MD), and H2O2. The topic is of interest and within the scopus of the journal, as C. albicans is a major fungal pathogen, capable of causing severe infections in humans. The manuscript is well organized, written in a compreensible way, and minor language corrections are required. There are a few questions which the authors should address:
1) In the manuscript, the authors examined the consequences of Wor1 overexpression on several traits of C. albicans physiology. However, nothing is mentioned concerning the consequences of the deletion of the gene. The analysis of the consequences of deleting a gene followed by its complementation is a more classical approach to demonstrate the function of a gene. At least some comments on the consequences of Wor1 deletion on the physiology and pathogenicity of Wor1 deletion should be mentioned.
line 61: wo in bold?
line 64-65: "GUT cells have a 65 different cell surface architecture to opaque cells...". GUT cells have a cell surface architecture different from opaque cells?
line 69: explain the meaning of WOR1OE, as it is the first time it appears.
line 91: drops were 10 microliter?
lines 133-134: a link to the table is missing.
Figure 1A: The color light blue is the purple on the legend?
line 218: The scale bar represents 10 µm cell size?
line 316: "but can be related with the increased tolerance to azoles displayed by WOR1OE strain". How? An explanation would help the reader.
Line 441: The conclusion that Wor1 overexpression is demonstrated as contributing to gut colonization is not supported by experimental evidences. Re-write.
Reviewer 3 Report
The article by Vico et al describes the role of Wor1 in regulating lipid metabolism, mitochondrial function, and antifungal drug resistance in fungal pathogen Candida albicans. Wor1 is a white opaque switching master regulator and the mechanism by which it regulates the switching is not well characterized. This article is very interesting. Here are my comments:
1) Was the overexpression of Wor1 in the constructed strain verified with qPCR? If so, please provide the data.
2)Why did the authors not check the expression of efflux pumps especially CDR1 as it is commonly associated with azole resistance? Please clarify. This is important especially when the authors found the expression of ERG11 to be decreased in WOR1 OE strain
3) Lines 133-134: Typing error
4)How were the qPCR data analyzed? Did the authors used delta-delta Ct method. IF so the authors should mention in the Materials and Methods. The authors should mention here that they used Actin as a house-keeping gene control.
5) Did the authors perform DNAse 1 step to remove the contaminating gDNA
6)Fig 4: The reviewer is curious as to why the authors did not use the hog1 deletion background strain as well for these assays. Please clarify
7) Fig 5d: Can the authors quantify the MFIs in flowcytometry while using the JC-1 dye? Please clarify
Reviewer 4 Report
In the paper entitled "Overexpression of the white opaque switching master regulator Wor1 alters lipid metabolism and mitochondrial function in Candida albicans", the authors are seeking the role of Wor1 in lipid homeostasis, mitochondrial function, azole susceptibility, and response to oxidative stress and finally show that Wor1 overexpression in C. albicans is altering the metabolism of the cell, connecting mitochondrial function, lipid metabolism and gut colonization.
In my opinion, this research is complex and valuable. The paper is well written (there are some exceptions though), well organized and therefore could be accepted for publication in JoF, after some minor changes:
· Minor English language editing;
· The Abstract is poorly written in English, and therefore is not clear or comprehensible enough; I advise the authors to rewrite this part;
· Online Supplementary Material cannot be accessed (at least by me...error 404, page not found);
· (Celsius) degree symbol must be replaced with the correct one: "°" (no underline);
· Page 4 - lines 133-134: wrong citing;
· Concerning 3.4. WOR1 overexpression alters the susceptibility to membrane disturbing compounds: the resistance to azoles (or even to allyamines, maybe) makes sense, but when we say "membrane disturbing compounds", we also include membrane function alternating polyene macrolides, whose activity might be enhanced in this case, considering the "thinner" membrane; it would be interesting to check this idea or maybe the authors have already investigated it; nonetheless, the title of this paragraph should be changed in order to have antifungal macrolides excluded;
· I believe the main text is written in 2 font sizes (check the first rows on pages 2 and 11).
Round 2
Reviewer 1 Report
My comments have been addressed and I have no further suggestions for improvement.
Author Response
We think the last version was clear enough to solve your doubts. Since another reviewer has requested other information we are submitting a new document with an extended version in the Discussion section.
Reviewer 2 Report
In the revised version of the manuscript, the main criticism raised and suggestion to the authors to address the consequences of WOR1 deletion were not considered and therefore the biological significance of the work remains unclear. At which point natural strains will overexpress the gene and what is the biologicla significance of this? Although minor issues were correctly addressed, my recommendation is the rejection of the manuscript.
Author Response
We have considered the suggestion of the reviewer and have discussed it in the Discussion section. As you may observed in the attached file, we still do not know if you are requesting additional work regarding to wor1 mutants which we consider not necessary as it is not the topic of the present work. We hope the new version and the reasons we give you (see the attached file) will answer the questions raised.

Reviewer 3 Report
The authors have satisfactorily answered to all my queries
Reviewer 4 Report
The authors improved a lot the quality of their article, and including amphotericine B in the study was a very pleasant surprise.